# Jacaric Acid Empowers RSL3-Induced Ferroptotic Cell Death in Two- and Three-Dimensional Breast Cancer Cell Models

**DOI:** 10.3390/ijms26073375

**Published:** 2025-04-04

**Authors:** Géraldine Cuvelier, Perrine Vermonden, Pauline Debisschop, Manon Martin, Françoise Derouane, Gerhard Liebisch, Josef Ecker, Marcus Hoering, Martine Berlière, Mieke Van Bockstal, Christine Galant, François Duhoux, Larissa Mourao, Colinda Scheele, Olivier Feron, René Rezsohazy, Cyril Corbet, Yvan Larondelle

**Affiliations:** 1Louvain Institute of Biomolecular Science and Technology, UCLouvain, 1348 Louvain-la-Neuve, Belgium; geraldine.cuvelier@uclouvain.be (G.C.); pauline.debisschop@uclouvain.be (P.D.); manon.martin@uclouvain.be (M.M.); rene.rezsohazy@uclouvain.be (R.R.); yvan.larondelle@uclouvain.be (Y.L.); 2Pole of Medical Imaging, Radiotherapy and Oncology (MIRO), Institut de Recherche Expérimentale et Clinique (IREC), UCLouvain, Avenue Hippocrate 57, 1200 Brussels, Belgium; francoise.derouane@uzleuven.be (F.D.); francois.duhoux@uclouvain.be (F.D.); 3Department of General Medical Oncology and Multidisciplinary Breast Center, Leuven Cancer Institute, University Hospitals Leuven, 3000 Leuven, Belgium; 4Institute of Clinical Chemistry and Laboratory Medicine, University Hospital Regensburg (UKR), 93053 Regensburg, Germany; gerhard.liebisch@klinik.uni-regensburg.de (G.L.); josef.ecker@ukr.de (J.E.); marcus.hoering@klinik.uni-regensburg.de (M.H.); 5Department of Gynecology, King Albert II Cancer Institute, Cliniques Universitaires Saint-Luc, Avenue Hippocrate 10, 1200 Brussels, Belgium; martine.berliere@uclouvain.be; 6Pole of Gynecology (GYNE), Institut de Recherche Expérimentale et Clinique (IREC), UCLouvain, Avenue Mounier 52, 1200 Brussels, Belgium; 7Department of Pathology, Cliniques Universitaires Saint-Luc, Avenue Hippocrate 10, 1200 Brussels, Belgium; mieke.vanbockstal@saintluc.uclouvain.be (M.V.B.); christine.galant@uclouvain.be (C.G.); 8Pole of Morphology (MORF), Institut de Recherche Expérimentale Et Clinique (IREC), UCLouvain, Avenue Mounier 52, 1200 Brussels, Belgium; 9Department of Medical Oncology, Institut Roi Albert II, Cliniques Universitaires Saint-Luc, Avenue Hippocrate 10, 1200 Brussels, Belgium; 10Laboratory for Intravital Imaging and Dynamics of Tumor Progression, VIB Center for Cancer Biology, KU Leuven, 3000 Leuven, Belgium; larissa.mourao@kuleuven.be (L.M.);; 11Department of Oncology, KU Leuven, 3000 Leuven, Belgium; 12Pole of Pharmacology and Therapeutics (FATH), Institut de Recherche Expérimentale et Clinique (IREC), UCLouvain, Avenue Hippocrate 57, B1.57.04, 1200 Brussels, Belgium; olivier.feron@uclouvain.be (O.F.); cyril.corbet@uclouvain.be (C.C.); 13WEL Research Institute, Avenue Pasteur 6, 1300 Wavre, Belgium

**Keywords:** ferroptosis, jacaric acid, conjugated linolenic acids, breast organoids, lipid peroxidation, cancer

## Abstract

Ferroptosis has recently emerged as a promising strategy to combat therapy-resistant cancers. As lipid peroxidation is a key trigger of ferroptotic cell death, enhancing cancer cell susceptibility through the supply of highly peroxidisable fatty acids represents a novel therapeutic approach. Conjugated linolenic acids (CLnAs) fulfill this requirement, exhibiting a peroxidation propagation rate eight times higher than their non-conjugated counterpart, α-linolenic acid. This study evaluates jacaric acid (JA), a plant-derived CLnA, as a ferroptotic inducer, both as a monotherapy and in combination with RAS-selective lethal 3 (RSL3), a canonical ferroptosis inducer, in 2D and 3D breast cancer cell models. JA treatment significantly reduced cell viability across all models, primarily through lipid peroxidation driven by JA incorporation into cellular lipids rather than alterations in anti-ferroptotic gene expression. Moreover, JA synergistically enhanced RSL3 cytotoxicity under 2D and several 3D conditions. Similar effects were observed with punicic acid, another plant-derived CLnA isomer. Our study exploits a common feature of cancer metabolism, increased fatty acid uptake, to turn it into a vulnerability. The incorporation of JA into breast cancer cells creates a highly peroxidisable environment that increases cancer cell sensitivity to RSL3, potentially reducing required doses and minimising side effects.

## 1. Introduction

Cancer cells exhibit altered lipid metabolism, characterized by increased de novo lipogenesis, fatty acid uptake, and oxidation, which support energy production and membrane synthesis for proliferation [1]. As dietary lipids influence the lipid composition of the tumour microenvironment (TME) [2], specific fatty acids may exert cytotoxic effects on cancer cells and enhance the efficacy of anti-cancer drugs [3]. For instance, Dierge et al. demonstrated the cytotoxic effect of long-chain Ω-3 and Ω-6 polyunsaturated fatty acids (PUFAs) in acidic TMEs, especially when combined with a lipid droplet biogenesis inhibitor [4].

Among PUFAs, conjugated linolenic acids (CLnAs) have emerged as promising anti-carcinogenic agents [5,6]. CLnAs are octadecatrienoic acid (C18:3) isomers featuring at least two conjugated double bonds [3]. Plant-derived CLnAs include seven isomers with variations in double bond positions and geometry. These fatty acids have gained scientific interest due to their anti-obesity, anti-inflammatory, and anti-carcinogenic properties, as well as their role in regulating lipid metabolism [7]. Their anti-cancer potential is linked to ferroptosis induction, as their high susceptibility to lipid peroxidation drives oxidative cell death [5,8]. Among CLnA isomers, jacaric acid (JA) and punicic acid (PunA) share a cis-trans-cis geometry but differ in double bond positioning (8, 10, 12 for JA; 9, 11, 13 for PunA) (Figure 1A). JA is the most cytotoxic CLnA toward cancer cells, while PunA is the most abundant, constituting up to 80% of fatty acids in pomegranate seed oil [9,10,11].

Ferroptosis, first defined in 2012, is a regulated form of cell death triggered by excessive lipid hydroperoxides [12]. Its relevance has expanded beyond cancer to neurodegenerative diseases and ischemia-reperfusion injuries [13]. Ferroptosis arises from an imbalance between pro- and antioxidant mechanisms. Key antioxidant systems include glutathione/glutathione peroxidase 4 (GSH/GPX4), ubiquinol/ferroptosis suppressor protein 1 (FSP1), and tetrahydrobiopterin/GTP cyclohydrolase (BH4/GCH1), all of which counteract lipid peroxidation [14,15,16]. Lipid peroxidation can be initiated either enzymatically, driven by lipoxygenases (LOX) and cytochrome P450 oxidoreductases (POR), which convert their substrate, PUFAs, into hydroperoxides, or non-enzymatically through autooxidation or oxidative reactions involving reactive oxygen species (ROS) [17,18,19]. Additionally, monounsaturated fatty acids (MUFAs) can confer resistance to lipid peroxidation by competing with peroxidisable PUFAs for phospholipid (PL) incorporation. The pro-oxidative status of cells is dictated by intracellular iron levels, which promote Fenton reactions and subsequent ROS production, and PUFA-enriched PLs [20]. PUFA incorporation into PLs requires acyl-CoA synthetase long-chain family member 4 (ACSL4) and lysophosphatidylcholine acyltransferase (LPCAT3) [21,22,23,24]. In 2017, Doll et al. demonstrated that ACSL4 expression correlates with ferroptosis sensitivity in breast cancer cell lines [25].

Breast cancer, due to its high prevalence and lipid-rich environment, has been a primary model for ferroptosis research. Triple-negative breast cancer (TNBC), the most aggressive subtype, is particularly sensitive to ferroptosis inducers [25]. High ACSL4 expression and estrogen receptor (ER) deficiency lead to increased incorporation of PUFAs into the phospholipid membrane and decreased abundance of cellular MUFAs. These two features contribute to TNBC’s sensitivity and distinguish it from luminal subtypes [25,26].

This study investigates the potential of a highly peroxidisable, plant-derived fatty acid to synergise with a canonical ferroptosis inducer, aiming to achieve two main objectives: first, to reduce the necessary dose of the canonical ferroptosis inducer, thereby minimising side effects, and second, to explore new therapeutic options for treatment-resistant tumours of the TNBC subtype. Additionally, we aim to gain a deeper understanding of how JA mediates its pro-ferroptotic effects. We examined JA-induced ferroptosis across three in vitro breast cancer models: (1) 2D-cultured breast cancer cell lines, including three luminal and four TNBC lines, selected to represent each subtype and assess potential differences in sensitivity to JA treatment; (2) 3D spheroid cultures; and (3) patient-derived organoids (PDOs). JA treatment reduced viability across all models. In 2D cultures, sensitivity to JA-induced ferroptosis correlated with intracellular lipid peroxidation, and with lipid profile changes in PL and neutral lipid (NL) fractions. Transcriptomic analysis showed no significant changes compared to untreated cells, suggesting that JA-induced ferroptosis is driven primarily by lipidome remodelling and lipid peroxidation rather than transcriptomic or enzymatic regulation. Finally, JA synergised with Ras-selective lethal 3 (RSL3), a pharmacological GPX4 inhibitor, in both 2D and 3D models, highlighting its therapeutic potential for future in vivo studies.

## 2. Results

### 2.1. JA Induces Ferroptotic Cell Death in Both Two- and Three-Dimensional-Cultured Breast Cancer Models

JA was tested at various concentrations on four TNBC and three luminal A cell lines cultured as monolayers (2D). All cell lines underwent complete loss of viability at a maximal concentration of 30 µM of JA or below (Figure 1B). With the exception of the T47D cell line, the sensitivity of breast cancer cell lines to JA-induced cell death corresponded with the sensitivity of their subtypes to the canonical ferroptotic inducers [25]. In contrast, α-linolenic acid (ALA), its non-conjugated counterpart, failed to induce cell death in Hs578T and MCF7 cells, chosen to represent the TNBC or luminal A subtype, respectively (Appendix A). When the same two cell lines were co-treated with a lethal concentration of JA and cell death inhibitors for apoptosis, necroptosis, and ferroptosis, only Fer1 managed to prevent cell death, confirming ferroptosis as the mechanism of JA-induced cell death (Figure 1C and Appendix A). Lipid peroxidation levels of the two cell lines treated with JA alone were increased compared to their respective control. This increase in lipid peroxidation levels appeared to correlate positively with the sensitivity of the cell lines to JA, which was higher in the TNBC Hs578T cell line and was rescued when Fer1 was added to the culture medium (Figure 1D). MCF7 and Hs578T cells were also cultured in ultra-low attachment (ULA) plates to form 3D spheroids. JA also induced a loss of viability of both cell lines in this 3D model, although higher concentrations were required to achieve cell death (Figure 1E). Compared to 2D-cell viability assays, the pattern of sensitivity to JA was maintained. Finally, eight PDOs, including two luminal PDOs and six TNBC PDOs, were treated with various concentrations of JA. Despite a higher threshold for JA-induced cell death compared to cell line-derived spheroids, JA significantly decreased the viability of organoids and did so more strongly when applied at high doses (i.e., ≥60 µM) (Figure 1F). One organoid model (IDC031) showed less than 50% viability when treated with 30 µM JA, three organoid models showed 50% or less viability when treated with 60 µM JA, and almost all organoid models showed less than 50% viability when treated with 100 µM JA (Figure 1F). These results emphasise the therapeutic potential of inducing ferroptosis with JA against breast cancer. When treating the above-mentioned in vitro BC models with PunA, similar results were obtained (Appendix A

### 2.2. JA Induces Changes in the Lipidome of Breast Cancer Cells, While Barely Affecting the Transcriptome

#### 2.2.1. JA Integrates into the PLs and NLs of Breast Cancer Cells

A fatty acid profile was performed in two TNBC breast cancer cell lines, Hs578T and MDA-MB-468, along with the luminal A cell line, MCF7, following a 2-h treatment with 25 µM of JA. As shown in Figure 2A, JA integrated into both the NL and PL fractions in all three cell lines. The PL fractions exhibited a notably higher integration of JA, with values ranging from 1.5-fold for Hs578T to approximately 4-fold for MDA-MB-468 and MCF7, in comparison to the NL fractions. Notably, no JA was detected in the FFA fractions. The ratio of MUFAs to PUFAs in both PL and NL fractions was also analysed (Figure 2B). MCF7 cells showed more than a two-fold higher MUFA/PUFA ratio in the PLs compared to the other two cell lines. Since MCF7 cells display, lower sensitivity to JA and MUFAs are potent inhibitors of ferroptosis, inhibiting stearoyl-CoA desaturase 1 (SCD1) could reduce MUFA abundance and increase MCF7 sensitivity to CLnA treatment. Accordingly, MCF7 cell viability was significantly reduced upon treatment with PunA and the SCD1 inhibitor, A939572. However, no such reduction was observed in Hs578T cells, which have a lower MUFA to PUFA ratio (Appendix A).

#### 2.2.2. JA Treatment Leads to an Increase in Highly Unsaturated Phosphatidylcholine (PC) and Phosphatidylethanolamine (PE) Lipid Species in Breast Cancer Cells

To gain deeper insights into the lipid composition of cells treated with JA, with a focus on key neutral and PL species, a lipidomic analysis was conducted on MCF7 and Hs578T cells treated with 10 µM JA. The results showed a significant increase in lipid species containing three to six double bonds across five of the 15 lipid classes analysed (Figure 2C). This finding strongly suggests that JA was integrated into lipid species from the PE, PC, and phosphatidylinositol (PI) classes of the PL membrane in both cell lines, as the observed increase was restricted to species with more than three double bonds. Additionally, in Hs578T cells, an increased abundance in lipid species was observed in two NL classes: triacylglycerols (TG) and cholesteryl esters (CE), all of which contained three or more double bonds. This is summarized in Appendix A, which resumes the significant changes in lipid species of Figure 2C. These findings align with the results shown in Figure 2A, where a more substantial integration of JA into the NL fraction was observed in Hs578T cells compared to MCF7 cells. No significant differences were found between the lipidomes of cells treated with JA and those treated with 10 µM PunA, suggesting that both CLnAs integrate into the same lipid classes with a similar distribution (Appendix A).

#### 2.2.3. Transcriptomic Analysis Reveals Minimal Differences Between JA-Treated and Control Cells

Transcriptomic analysis of cells treated with 3 µM of JA for 6 and 24 h revealed no significant differences in RNA expression between treated and untreated breast cancer cells (Figure 2D). Similarly, protein levels of key ferroptosis-related enzymes, including FSP1, GPX4, and DHFR, were unaffected by JA, PunA, or ALA treatment (Figure 2E).

These results suggest that CLnA-induced ferroptosis primarily arises from lipidome changes, leading to intense and uncontrolled lipid peroxidation rather than alterations in the expression of pro- or anti-ferroptotic enzymes. Furthermore, the ratio of PUFAs to MUFAs appears to influence the extent of lipid peroxidation, with a higher PUFA/MUFA ratio correlating with more severe lipid peroxidation.

### 2.3. JA Empowers the Efficacy of Ferroptosis Inducers on Breast Cancer Cells by Decreasing the Dose-Efficient Concentrations of RSL3

RSL3 is a well-known inducer of ferroptosis widely used to trigger ferroptosis in cell culture systems and act on GPX4. It covalently binds to the selenocysteine residue on the active site of GPX4, blocking its detoxifying function [27,28]. In the present study, RSL3 was used either as a single treatment or in combination with a sublethal concentration of JA to assess cell viability in both 2D and 3D breast cancer models. The LD50 of RSL3, when applied alone, was respectively 0.44, 9.56, and 9.79 µM for MDA-MB-468, Hs578T, and MCF7 cell lines grown in 2D (Figure 3A). In combination with a sublethal dose of JA, 0.3, 1, and 1.57 µM for respectively MDA-MB-468, Hs578T, and MCF7 cells, the LD50 of RSL3 dropped to 0.16, 0.21, and 0.06 µM for the same cell lines (Figure 3B). The drop was thus much more significant for the cell lines that are natively less sensitive to RSL3. These comparisons should, however, be considered with caution since the sublethal concentrations of JA used differed between cell lines. In 3D-spheroids, the same trend was observed as in 2D-cultured models: the viability of MDA-MB-468 spheroids treated with RSL3 in combination with JA was significantly lower than that of similar spheroids treated with RSL3 or JA alone (Figure 3C and Appendix A). For Hs578T and MCF7 cells, an almost complete loss in viability was observed when spheroids were treated with both RSL3 and JA. However, as RSL3 alone already induced an important decrease in viability, the amplifying effect of JA on RSL3 response was less obvious than for MDA-MB-468 spheroids. Concerning breast cancer organoids, JA applied as a single treatment at a dose of 60 µM had a variable cytotoxic potential on the different cell lines, ranging from a 27 to 58% decrease in viability. The same could be observed when treating organoids with RSL3 at a dose of 10 µM. The mean decreases in viability with RSL3 ranged between 0 and 54%. With the exception of IDC113 and IDC117 organoids, the sensitivity to JA coincided with the sensitivity towards RSL3. When applying both RSL3 and JA at the same time, the viability of all organoids decreased more significantly, with a mean decrease in viability ranging from 67 to 100%. For IDC113 and IDC117 organoids, the combinatory therapy especially appeared to strongly amplify individual effects as the decrease in viability was up to 40% higher than what would have been obtained for an additive effect (Figure 3D). Again, very similar trends could be observed when the different breast cancer models were co-treated with RSL3 and PunA (Appendix A).

## 3. Discussion

Ferroptosis is an emerging therapeutic strategy that exploits metabolic vulnerabilities in tumour cells. This study demonstrates the potent ferroptosis-inducing effects of JA, a CLnA isomer, across multiple breast cancer models. JA exhibits strong cytotoxicity in 2D-cultured breast cancer cells, regardless of subtype, with potency comparable to or exceeding established ferroptosis inducers. Consistent with prior research [8,29], JA is more effective than its geometric isomer, PunA, in triggering ferroptotic cell death.

JA efficacy extends to more physiologically relevant 3D breast cancer spheroids and PDOs, demonstrating its translational potential. While higher JA concentrations were required to achieve complete cell death in these models, this aligns with the increased resistance typically observed in 3D systems. Mechanistically, lipidomic analyses reveal that JA integrates into key lipid classes associated with ferroptosis sensitivity, promoting lipid peroxidation—a hallmark of ferroptotic cell death—without significant transcriptomic alterations. These findings highlight lipid remodelling as a central mechanism in JA-induced ferroptosis.

JA also exhibits a synergistic effect when combined with RSL3, a GPX4 inhibitor. This combination lowers RSL3 LD50 across diverse models, including PDOs, supporting its potential to enhance ferroptosis-based therapies.

While these results are promising, further studies are needed to assess JA pharmacokinetics, bioavailability, and therapeutic index. Prior research on CLnA-rich oils suggests favourable safety profiles [30], but additional studies are required to determine JA-specific toxicity in vitro and metabolism in vivo. Although JA cytotoxicity has already been successfully tested on differentiated Caco2 cells [31], immortalized epithelial breast cells, such as MCF12A, should be tested to assess their sensitivity to both JA and RSL3, either alone or in combination.

PDOs provide valuable insights into drug response, yet their lack of stromal and immune components limits their ability to fully replicate tumour complexity. Future research should integrate advanced co-culture models incorporating cancer-associated fibroblasts (CAFs) or peripheral blood lymphocytes that preserve native TME elements to better understand JA therapeutic potential [32,33,34,35,36]. Additionally, nanodelivery systems could improve JA bioactivity and stability in vivo, as JA and other CLnA isomers have been shown to be rapidly metabolized into less cytotoxic CLAs after absorption [37,38]. This would facilitate its translation into clinical applications [39,40,41,42].

In summary, JA demonstrates strong ferroptosis-inducing potential and enhances the effects of the ferroptosis inducer RSL3. While challenges related to its metabolism and therapeutic index remain, this study provides a foundation for future investigations aimed at optimizing JA-based ferroptosis therapies for clinical use.

## 4. Materials and Methods

### 4.1. Cell Lines, Organoid Models, and Culture Conditions

All breast cancer cell lines were obtained from ATCC and were cultured in RPMI1640 medium, glutaMAX (Gibco, Grand Island, NY, USA, 61870036) supplemented with 10% heat-inactivated fetal bovine serum (FBS, Gibco, A5256701) and 1% (*v*/*v*) penicillin/streptomycin (P/S) (Gibco, 15140122) in an atmosphere of 95% air/5% CO2 at 37 °C. Negative results for mycoplasma contamination were obtained with the MycoAlert detection kit (Lonza, Basel, Switzerland) prior to the study. BCO017, BCO018, and BCO019 organoid lines were established from biopsies of breast cancer patients from the Cliniques Universitaires Saint-Luc, Brussels, Belgium (ethical agreement UCL-ONCO2015-02—2015/13AOU/445). IDC031, IDC113, IDC117, IDC143, and IDC157 breast organoid lines were initiated from freshly resected tumor tissues obtained from breast cancer patients at Antoni van Leeuwenhoek Hospital, Amsterdam, The Netherlands (ethical agreement NKI-B17PRE). Organoids were grown in DMEM/F12 (Gibco, 21041025) supplemented with 1% P/S, 1% HEPES, and 1% glutamine, and completed with B27 and 16 growth factors as previously described [43]. The features of the different cell lines used are shown in Table 1.

### 4.2. Chemicals

JA (10-1876), PunA (10-1875), and α-linolenic acid (ALA, 10-1803) were purchased from Larodan (Solna, Sweden). All fatty acids were conjugated to fatty acid-free albumin in a 4:1 (*w*/*w*) ratio and dissolved in phosphate buffer saline (PBS) before being stored at −80 °C. Ferrostatin-1 (Fer1, SML0583), α-tocopherol (vitE, 3251-5G), and deferoxamine mesylate (DFO, D9533) were purchased from Sigma Aldrich (St. Louis, MO, USA). Necrostatin-1 (s8037), RSL3 (S8155), and ZVAD-FMK (S7023) were purchased from Selleck Chemicals (Houston, TX, USA). The stearoyl-CoA desaturase1 (SCD1) inhibitor, A939572, was purchased from Med-ChemExpress (Monmouth Junction, NJ, USA). With the exception of vitE, which was dissolved in ethanol, all other drugs were dissolved in DMSO.

### 4.3. Spheroid Model

Cells were seeded at a density of 1000 cells per well in 96-well ULA plates (Greiner, Kremsmünster, Austria, 650970) [44]. Each plate was spined for 5 min at 200× *g*; spheroids were allowed to grow for 3 days before treatment.

### 4.4. Organoid Model

Maintenance was performed using 24-well plates (Corning, NY, USA, 3524) at a density of 400,000 cells per well, embedded in Cultrex Basement Membrane Extract (BME) (Bio-Techne, Minneapolis, MN, USA, 3432-010-01), an organoid growth scaffold. Organoids were allowed to grow for one week between two passages. Up-and-down pipetting was performed during harvesting to destroy the dome of Cultrex containing the organoids. The organoid suspension was then centrifuged at 400× *g* for 5 min. After removing the supernatant, the pellet was suspended in 2 mL of trypsin-EDTA (Gibco, 25200056) and incubated at 37 °C for 15 to 30 min, depending on how easily organoids dissociated. Up-and-down pipetting was performed before incubation using a pipette P200 to facilitate single-cell obtention. Nine milliliters of PBS were added to neutralize trypsinization, and the tube was centrifuged to obtain a cell pellet. The pellet was resuspended, counted, and diluted in the required amount of ice-cold Cultrex BME in order to seed a volume of 40 µL per well. The plate was left for 40 min in a 37 °C incubator to allow the Cultrex BME to polymerize before adding 1 mL of culture medium to each well.

### 4.5. Cell Viability Assessment

Cells grown in 2D were seeded at a density of 10,000 cells per well in a 96-well plate and treated 24 h later with different concentrations of JA, PunA, or RSL3 ± JA or PunA. Cell viability was measured 72 h later using PrestoblueTM cell viability reagent (Fisher Scientific, Waltham, MA, USA, 12083745) according to the manufacturer’s instructions.

Cells grown as spheroids were treated with six different concentrations of PunA or JA or with a sublethal dose of CLnA, specific to each cell line, ±3 µM of RSL3 and renewed after 2 days. Cell viability was assessed directly in the 96-well ULA plate used to obtain the spheroids. Spheroid diameters were measured before each treatment with a microscope camera (Moticam 3.0 MP, Xiamen, China) with fixed exposure and gain settings. 20 µL of pure PrestoblueTM were added to the 200 µL in each well one week after cell seeding, and cell viability was measured 24 h later at an excitation/emission wavelength of 560/590 nm using Fluoroscan Ascent FL microplate fluorometer (Fisher Scientific).

Viability tests on organoids were performed using 96-well plates coated with 20 µL of Cultrex BME. Organoids were seeded at 10,000 cells per well in a culture medium supplemented with 2% of Cultrex BME and left to grow for 1 week before treatment. Cells were treated with JA alone at 0, 30, 60 and 100 µM or with 10, 30, 60 µM of JA or PunA ± 1, 3, 10 µM of RSL3. Unlike for spheroids, treatment was not renewed and left for one week before cell viability assessment using CellTiter-Glo (Promega, Madison, WI, USA, G7570) according to the manufacturer’s instructions with some modifications. Briefly, the lysis procedure was obtained through up-and-down pipetting before being transferred, after a 30-min incubation, into a Promega luminescent reader-compatible 96-well plate. The luminescence was read with a GloMax microplate reader (Promega).

### 4.6. Measurement of Lipid Peroxidation

Cells were seeded at a density of 15,000 cells per well in 96-well black/clear plates (Greiner, 655090). Treatment was performed 24 h after seeding with 25 µM of JA or PunA ± 10 µM of the ferroptosis inhibitor Fer1. After 6 h, wells were washed with PBS and 3 µM of the lipid peroxidation sensor C11-BODIPY 581/591 (4,4-difluoro-5-(4-phenyl-1,3-butadienyl)-4bora-3a,4a-diaza-s-indacene-3-undecanoic acid, Fisher Scientific, D3861), which is oxidized in the presence of lipid ROS, were added to each well. The plate was protected from light and incubated for 30 min in the incubator. Fluorescence was read with the Tecan Infinite Mpro200 device. The red fluorescence (excitation = 580 nm, emission = 620 nm) corresponded to the non-oxidized probe, and the green fluorescence (excitation = 500 nm, emission = 540 nm) corresponded to the oxidized probe. Relative fluorescence, calculated upon division of the green by the red fluorescence, was used to quantify relative intracellular lipid peroxidation levels. The relative fluorescence of all conditions was divided by the control condition to obtain a fold change value.

### 4.7. Immunoblotting

Cells were cultured for 24 h in 6-well plates at a seeding density of 300,000 cells per well for 2D-cultured cancer cells. Cells were then treated for 24 h with 1 µM of either JA, PunA or ALA before harvest and lysis with home-made RIPA buffer consisting of 50 mM Tris-HCl at pH 7.4, 150 mM NaCl, 1 mM EDTA, 1% *v*/*v* Triton X, 0.05% *w*/*v* sodium deoxycholate, 1% *w*/*v* SDS and 1% *v*/*v* of protease inhibitor cocktail (Sigma-Aldrich, St. Louis, MO, USA, P8340). A western blot was then conducted using the same method as previously described [31]. The following primary and secondary antibodies were used for immunoblotting: anti-ACSL4 (Abcam, Cambridge, UK, ab264397, 1:1000), anti-DHFR (Cell Signaling, Danvers, MA, USA, 45710, 1:1000), anti-Ecad (Cell Signalling, 14472, 1:1000), anti-ERα (Bioké, Leiden, The Netherlands, 13258S, 1:1000), anti-FSP1 (Santa Cruz, Dallas, TX, USA, 377120, 1:1000), anti-GPX4 (Abcam, ab125066, 1:1000), anti-β-Actin (Merck, Darmstadt, Germany, A5441, 1:10,000), anti-VCP (Cell signaling, 2648, 1:1000), HRP-linked horse anti-mouse IgG (Cell Signaling, 7079, 1:5000), HRP-linked goat anti-rabbit IgG (Cell signaling, 7074, 1:5000) and HRP-linked goat anti-rat IgG (Cell signaling, 7077, 1:5000). All primary and secondary antibodies were diluted in 5% and 1% *w*/*v* respectively skimmed milk in TBS-0.1% *v*/*v* Tween 20 (TTBS).

### 4.8. Lipidomic Analysis and Fatty Acid Analysis by Gas Chromatography

Prior to lipidomic analysis, cells were treated with 10 µM of JA or PunA, and treatment was left for 4 h. Following Bligh and Dyer extraction, lipidomic analysis was performed on total cellular lipid extracts by liquid chromatography followed by Fourier-transform mass spectrometry as previously reported [45]. All analyses were performed in R (csv4.2.2) [46]. Lipid species with a frequency of 70% missing values were filtered out, and lipid species abundances were log2 transformed. In order to identify the lipid species whose abundance changes with treatments, differential expression analysis (DEA) was conducted on the abundances of all lipid species for all cell lines by performing a regression with the treatment effect based on the proDA package (version 1.10.0) [47]. This method accounts for missing values without imputation by implementing a probabilistic dropout model. Significance was then established with a Wald test on the model coefficient of interest, and *p*-values were adjusted using the False Discovery Rate for multiple tests.

To assess the amount of JA integrated into the NLs and PLs of MDA-MB-468, Hs578T, and MCF7 cells, cells were treated with 25 µM of JA and harvested 4 h after treatment. A Bligh and Dyer extraction, followed by solid phase extraction, fatty acid methylation, and gas chromatography analysis, was performed as previously described [5].

### 4.9. RNA Sequencing

MCF7 and Hs578T cells were cultured in 6-well plates at a seeding density of 300,000 cells per well. Three µM of JA were applied to the cells 24 h after seeding, and cells were harvested after 6 and 24 h to test two different time points in three independent biological replicates. Total RNA was extracted using the RNeasy Mini Kit (Qiagen, Hilden, Germany, 74104) according to the manufacturer’s instructions. The quality of the RNA was tested using the Qubit RNA IQ assay kit (Invitrogen, Waltham, MA, USA, Q10210). Total RNA was then sent to Macrogen to perform the stranded mRNA-seq library construction and sequencing using the Illumina NovaSeq 6000 platform (San Diego, CA, USA)(150 base pairs paired-end, with a sequencing depth of 30 million reads per sample). Macrogen further confirmed the RNA integrity and purity with the Agilent 2100 BioAnalyzer (Santa Clara, CA, USA). All samples met the Macrogen requirements of library construction and were submitted for RNA sequencing. Raw RNA sequencing reads in FASTQ format were processed using a standard RNAseq pipeline, including Trimmomatic (v0.39) [48] to remove low-quality reads, HISAT2 (v2.2.1) [49] to align reads to the human genome (GRCh38), and gene expression levels were evaluated using featureCounts from Subread (v2.0.3) [50] and Homo_sapiens.GRCh38.105.chr.gtf. Genes for which raw counts were null for all samples were filtered out. The DESeq2 R package was used to determine differentially expressed genes from raw counts in different conditions [51].

### 4.10. Statistics

All results are expressed as the mean ± the standard error of the mean (SEM) or the standard deviation (SD), as indicated in the legend of the figures. Except for lipidomic and transcriptomic analyses for which statistical analysis was performed with R (see above), statistics were performed with GraphPad Prism 9.1 and performing one-way ANOVA with Tukey’s multiple comparisons, two-way ANOVA with Sidak’s multiple comparisons or Kruskal-Wallis test with Dunn’s multiple comparisons. Statistical significance to the control or to another treatment is determined as follows: * *p* ≤ 0.05, ** *p* ≤ 0.01, *** *p* ≤ 0.001.

## Figures and Tables

**Figure 1 ijms-26-03375-f001:**
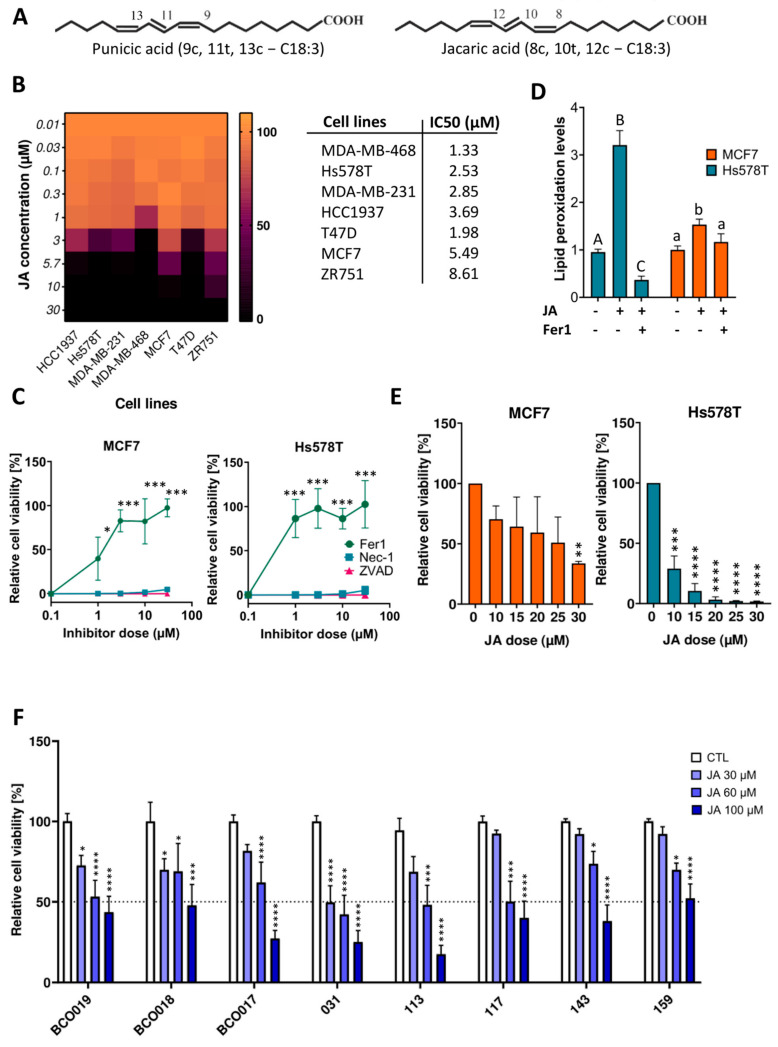
Jacaric acid (JA) induces ferroptotic cell death in both two- and three-dimensional-cultured breast cancer models. (**A**) Chemical structures of JA and punicic acid (PunA). (**B**) Relative viability of MDA−MB−231, MDA−MB−468, HCC1937, Hs578T, T47D, MCF7, and ZR751 cells after 72h of treatment with 5 different doses of JA, normalized to the control (untreated cells). The scale on the right side of the heatmap represents viability as a percentage. Associated LD50 values of graph B are shown in the table on the right of the heatmap. (**C**) Relative viability of Hs578T and MCF7 cells after 72h of treatment with a lethal dose of JA, namely 25 µM, and increasing concentrations of ferroptosis, necroptosis, and apoptosis inhibitors, namely ferrostatin−1 (Fer1), necrostatin-1 (Nec−1) and Z−VAD−FMK (ZVAD), normalized to the control (untreated cells). (**D**) Lipid peroxidation levels (determined as the fold change in the green-to-red fluorescence ratio) in Hs578T and MCF7 cells treated with 25 µM of JA ± 10 µM of Fer1 for 6 h compared to the control (untreated cells). (**E**) Relative viability of Hs578T- and MCF7-derived spheroids treated for 5 days with 6 different concentrations of JA, normalized to the control (untreated cells). (**F**) Relative viability of BCO017, BCO018, BCO019, IDC031, IDC113, IDC117, IDC143, and IDC159 organoids treated with 4 different concentrations of JA for 1 week, normalized to the control (untreated cells). Results are expressed as mean ± standard error of the mean (**D**,**F**), as mean ± standard deviation (**A**–**C**,**E**) of three independent repetitions. Significance was established by one-way or two-way ANOVA with Dunnett’s multiple comparisons (**C**,**E**,**F**) or by the Kruskal-Wallis test with Dunn’s multiple comparisons (**D**). Statistical significance is indicated by letters (**D**) and asterisks (**C**,**E**,**F**): capital letters (A–C) are used for comparisons of treatments in the Hs578T cell line, while lowercase letters (a,b) are used for comparisons of treatments in the MCF7 cell line. Asterisks denote significance levels as follows: * *p* < 0.05, ** *p* < 0.01, *** *p* < 0.001, **** *p* < 0.0001.

**Figure 2 ijms-26-03375-f002:**
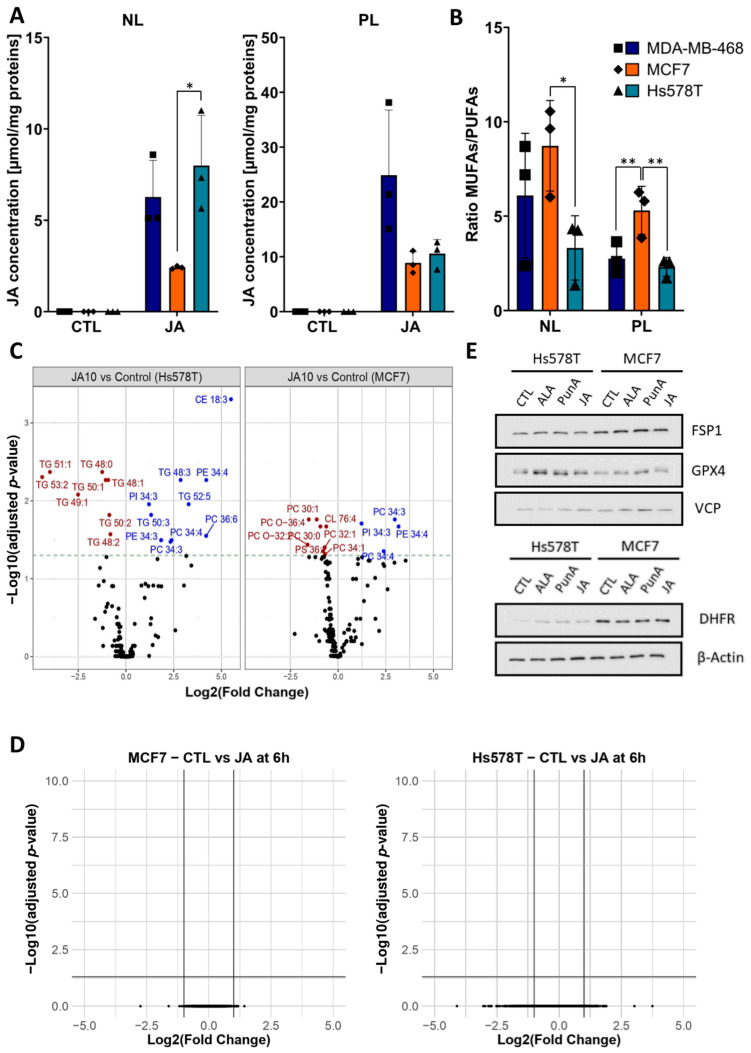
Jacaric acid (JA) induces changes in the lipidome of breast cancer cells while barely affecting the transcriptome. (**A**) Bar charts showing the content of JA (µmol/mg of protein) in the neutral lipid (NL) and phospholipid (PL) fractions of Hs578T, MDA-MB-468, and MCF7 cells upon treatment with 25 µM of JA for 2 h. Each symbol represents a distinct biological replicate, each cell line being identified by a symbol of a unique shape. No asterisk indicates a non-significant (n.s.) difference. (**B**) Bar plots showing the ratio of monounsaturated fatty acids (MUFAs) to polyunsaturated fatty acids (PUFAs) in the NL and PL of the MDA-MB-468, Hs578T, and MCF7 cells. Each symbol represents a distinct biological replicate, each cell line being identified by a symbol of a unique shape. No asterisk indicates a non-significant (n.s.) difference. (**C**) Volcano plots of significance (-log2 of adjusted *p*-value) and log2 fold change in the abundance of lipid species for Hs578T and MCF7 cells after treatment with 10 µM of JA for 4 h compared to the control (untreated cells). The 16 lipid species classes analysed were the followings: free cholesterol (FC), phosphatidylcholine (PC), phosphatidylethanolamine (PE), phosphatidylinositol (PI), phosphatidylserine (PS), phosphoglycerides (PG), triacylglycerol (TG), diacylglycerol (DAG), ceramides (Cer), hexosylceramides (hex_Cer), lysophosphatidylcholine (LPC), lysophosphatidylethanolamine (LPE), phosphatidylcholine ether lipids (PC_O), phosphatidylethanolamine ether lipids (PE_O), sphingomyeline (SM), cholesteryl esters (CE). (**D**) Volcano plots of significance (-log10 of adjusted *p*-value) and log2 fold change for the RNA transcripts of Hs578T and MCF7 cells after treatment with 1 µM of JA for 6 h. (**E**) Immunoblots of the ferroptosis-related proteins, ferroptosis suppressor protein 1 (FSP1), glutathione peroxidase 4 (GPX4) and dihydrofolate reductase (DHFR), and of the reference proteins, β-Actin and Valosin-containing protein (VCP) in Hs578T and MCF7 cells after 24 h of treatment with 1 µM of α-linolenic acid (ALA), punicic acid (PunA) or JA. Results are expressed as mean ± standard deviation of three independent repetitions (**A**,**B**). Immunoblots are representative of three independent repetitions. Significance was established by one-way ANOVA with Tukey’s multiple comparisons (**A**), by two-way ANOVA with Sidak’s multiple comparisons (**B**), or by false-discovery rate with Benjamini-Hochberg *p*-value adjustment for multiple comparisons (**C**,**D**). * *p* < 0.05, ** *p* < 0.01.

**Figure 3 ijms-26-03375-f003:**
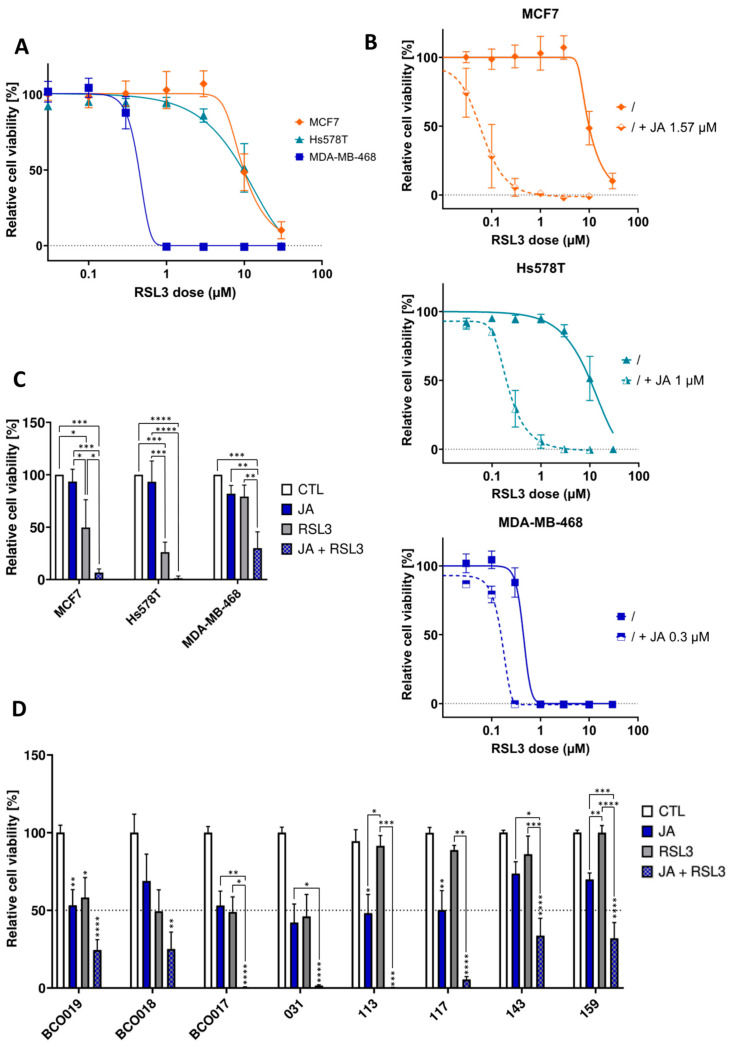
Jacaric acid (JA) empowers the efficacy of ferroptosis inducers on breast cancer cells by decreasing the dose-efficient concentrations of Ras-selective lethal 3 (RSL3). (**A**) Relative viability of MDA-MB-468, Hs578T, and MCF7 cells treated during 72 h with various concentrations of RSL3, normalized to the control (untreated cells). (**B**) Relative viability of MDA-MB-468, Hs578T, and MCF7 cells treated during 72 h either with different concentrations of RSL3, either alone or in combination with a fixed concentration of JA, normalized to the control (untreated cells). Dose response curves have been fitted to the data. (**C**) Relative viability of spheroids treated for 5 days with 1 µM of JA for Hs578T and MDA-MB-468 cells and 3 µM of JA for MCF7 cells ± 1 µM of RSL3. (**D**) Relative viability of BCO017, BCO018, BCO019, IDC031, IDC113, IDC117, IDC143, and IDC159 breast organoids after 1 week of treatment with 60 µM of JA or 10 µM of RSL3 or a combination of both, normalized to the control (untreated cells). Results are expressed as mean ± standard error of the mean (**D**) or as mean ± standard deviation (**A**–**C**) of three independent repetitions. Significance was established by one-way ANOVA with Tukey’s multiple comparisons (**C**,**D**) or by the Kruskal-Wallis test with Dunn’s multiple comparisons (**D**). * *p* < 0.05, ** *p* < 0.01, *** *p* < 0.001, **** *p* < 0.0001.

**Table 1 ijms-26-03375-t001:** Cell line and organoid subtypes.

Cell/Organoid Lines	Phenotype	Tumour Origin
BCO-17	TNBC	Primary tumour
BCO-18	HR + HER2+	Primary tumour
BCO-19	HR+	Primary tumour
IDC031	TNBC	Primary tumour
IDC113	TNBC	Primary tumour
IDC117	TNBC	Primary tumour
IDC143	TNBC	Primary tumour
IDC159A	TNBC	Primary tumour
HCC1937	TNBC	Primary tumour
MDA-MB-468	TNBC	Metastasis
MDA-MB-231	TNBC	Primary tumour
Hs578T	TNBC	Primary tumour
MCF7	Luminal A	Metastasis
T47D	Luminal A	Metastasis
ZR751	Luminal A	Primary tumour

## Data Availability

The original data presented in the study are openly available in Dataverse at https://doi.org/10.14428/DVN/HF01QB and https://doi.org/10.14428/DVN/NXEYJ6.

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
