# Peer review of "Jacaric Acid Empowers RSL3-Induced Ferroptotic Cell Death in Two- and Three-Dimensional Breast Cancer Cell Models"

_ijms, 2025, doi:10.3390/ijms26073375_

Round 1
Reviewer 1 Report
Comments and Suggestions for Authors
In this work, the authors study the response to jacaric acid (JA), a conjugated linolenic acid of natural origin, on the survival of breast cancer cells in culture. This compound integrates into biological membranes where it is capable of inducing cell death by ferroptosis, which could form the basis of new anti-cancer strategies.
The main result suggests that the higher the percentage of polyunsaturated fatty acids, the greater the response to JA. However, a study on a larger collection of cell lines would be needed to really validate this hypothesis.
Overall, this paper is based on relevant experimental approaches, albeit limited in scope due to the small number of cell types studied.
However, the manuscript suffers from too many shortcomings.
Below are a number of comments that may help the authors to improve their manuscript.
Major points:
- The most important part of the manuscript is devoted to lipidomic analyses. Unfortunately, it is also the most confusing part. In this respect, paragraph 2.2 needs to be revised, and would benefit from being divided into sub-sections.
Figure 1B. Please explain in the legend what the vertical scale on the right corresponds to.
Figure 2A-B and Figure S2A. The meaning of the symbols in black (triangles, squares, diamonds) should be explained in the figure legend.
Figure 2A-B. On histograms, please use the symbol n.s. to highlight results that are non-statistically significant.
Figure 2C. Do the results correspond to lipid quantification (as indicated in the figure legend, lines 249-50) or to a measurement of the presence of JA in the corresponding lipid fractions (as indicated in the results section, lines 225-226)? Please clarify this point. The same applies to Figure 2D.
Figure 2C and Figure S2D. It seems that these two figures are two different representations of the same results. Please clarify this point in the text.
Figure S2A. In this form, the figure is not very informative. It should be replaced by a table.
Minor points. The use of acronyms should be harmonised. It appears that the authors use both TAG and TG to denote triacylglycerol compounds. Also, "CE" does not appear in the list of acronyms. Minor points. The use of acronyms should be harmonised. It appears that the authors use both TAG and TG to denote triacylglycerol compounds. Also, "CE" does not appear in the list of acronyms.
Author Response
We greatly appreciate your insightful comments and the time you dedicated to reviewing our manuscript. Your feedback has been very helpful, and we have carefully considered each point. We’ve made the necessary revisions to address your suggestions and improve the overall quality of the manuscript. Below, you will find our responses to the specific comments. Thank you again for your valuable input.
Comments 1: The most important part of the manuscript is devoted to lipidomic analyses. Unfortunately, it is also the most confusing part. In this respect, paragraph 2.2 needs to be revised, and would benefit from being divided into sub-sections.
Response 1: Thank you for this suggestion. We have revised paragraph 2.2 by restructuring it into sub-sections to improve clarity and readability. (line 173-217)
Comments 2: Figure 1B. Please explain in the legend what the vertical scale on the right corresponds to.
Response 2: We have updated the legend of Figure 1B to clearly specify what the vertical scale on the right represents (i.e. The scale on the right side of the heatmap represents viability as a percentage.). (line 157-158)
Comments 3: Figure 2A-B and Figure S2A. The meaning of the symbols in black (triangles, squares, diamonds) should be explained in the figure legend.
Response 3: We have added explanations in the figure legends for Figures 2A-B to clarify the meaning of the black symbols. (line 222-223 and 225-226)
Comments 4: Figure 2A-B. On histograms, please use the symbol n.s. to highlight results that are non-statistically significant.
Response 4: To prevent overcrowding the figure, 'n.s.' was not added directly to it. Instead, the legend now explicitly states that the absence of asterisks in the figures indicates non-significant results. (line 223 and 226)
Comments 5: Figure 2C. Do the results correspond to lipid quantification (as indicated in the figure legend, lines 249-50) or to a measurement of the presence of JA in the corresponding lipid fractions (as indicated in the results section, lines 225-226)? Please clarify this point. The same applies to Figure 2D.
Response 5: We have now clearly stated that the data represent lipid quantification in both the figure legend and in the result section. The same has been done regarding the transcriptomic analysis represented in figure 2D. (line 194-195 and 227-229)
Comments 6: Figure 2C and Figure S2D. It seems that these two figures are two different representations of the same results. Please clarify this point in the text.
Response 6: We have revised the manuscript to explicitly state that Figures 2C and S2C represent different visualizations of the same data. (line 201-202)
Comments 7: Figure S2A. In this form, the figure is not very informative. It should be replaced by a table.
Response 7 : We have removed Supplementary Figure 2A, as it does not provide meaningful information. None of the cell lines incorporate JA into the free fatty acid (FFA) fraction, making the figure redundant.
Comments 8 : The use of acronyms should be harmonised. It appears that the authors use both TAG and TG to denote triacylglycerol compounds. Also, "CE" does not appear in the list of acronyms.
Response 8 : We have standardized the use of acronyms throughout the manuscript, consistently using TG. Additionally, we have added "CE" to the list of acronyms for clarity. (line 234)
Reviewer 2 Report
Comments and Suggestions for Authors
The article is not acceptable.
Considerations.
1. The title is inadequate. It does not present preclinical models of breast cancer. It should be changed to be more specific: As it says in the introduction:
“This study evaluates jacaric acid (JA), a plant-derived CLnA …. as a monotherapy and in combination with RAS-selective lethal 3 (RSL3)… in 2D and 3D breast cancer cell conditions”.
- It is worrying that if they are only cell studies, the article contains 18 authors.
- The introduction is too long, does not emphasize the importance of the study, and does not justify the use of so many cell lines.
- The significance of this study lies in understanding the adjuvant effects of various ferroptosis inducers. However, it is inconsistent that a normal cell line, such as MCF12, was not analyzed. Including this cell line would enhance the ongoing discussion about the benefits of using these components and their potential to reduce side effects.
- They wrongly mention that their 3D cultures are spontaneous spheroids since the low adhesion conditions induce the formation of these spheroids. The use of low adhesion conditions with medium without differentiation factors is a method for the enrichment of stem cells, which are effectively more resistant to different treatments. This is not the case since they only used low-adhesion dishes. What is the advantage of only using low adhesion?
- The results should be grouped according to cell type and differential responses. As they are now, this is confusing and irrelevant since all cell lines respond practically the same way.
- Discussion: Extremely repetitive and speculative. This study cannot support a section discussing side effects and immune modulation of the microenvironment.
Author Response
Thank you very much for your constructive comments on our manuscript. We have carefully considered all your suggestions and made the necessary revisions to improve the clarity and quality of our work.
Generally speaking, we acknowledge that many questions remain unanswered, but addressing them is beyond the scope of this work. Below, you’ll find our detailed responses to each of your points.
We hope the changes we’ve made address your concerns, and we appreciate your time and valuable feedback.
Comments 1 : The title is inadequate. It does not present preclinical models of breast cancer. It should be changed to be more specific: As it says in the introduction:
“This study evaluates jacaric acid (JA), a plant-derived CLnA …. as a monotherapy and in combination with RAS-selective lethal 3 (RSL3)… in 2D and 3D breast cancer cell conditions”.
Response 1 : The title has been revised to more accurately reflect the scope and content of the study. (line 2-3)
Comments 2 : It is worrying that if they are only cell studies, the article contains 18 authors.
Response 2 : The number of authors reflects the collaborative nature of this study, particularly due to the substantial effort required to generate and characterize the organoid models. As specifically detailed in the author contributions, several of these individuals were responsible for establishing the organoids (methodology) and providing them to us (resources).
Comments 3 : The introduction is too long, does not emphasize the importance of the study, and does not justify the use of so many cell lines.
Response 3 : The introduction has been shortened and the final paragraph of the introduction has been revised to better highlight the significance of the study and to provide a clearer justification for the inclusion of multiple cell lines. (line 104-113)
Comments 4 : The significance of this study lies in understanding the adjuvant effects of various ferroptosis inducers. However, it is inconsistent that a normal cell line, such as MCF12, was not analysed. Including this cell line would enhance the ongoing discussion about the benefits of using these components and their potential to reduce side effects.
Response 4 : The inclusion of an immortalized normal cell line like MCF12 would indeed provide valuable insights into the safety profile of our treatment on non-cancerous cells. This point has been further developed in the discussion. (line 314-315) However, these immortalized cell lines are still immortalized and do not provide full insight, as they proliferate much faster and are more sensitive to ferroptosis (see reference).
Rodencal J, Kim N, He A, Li VL, Lange M, He J, Tarangelo A, Schafer ZT, Olzmann JA, Long JZ, Sage J, Dixon SJ. Sensitization of cancer cells to ferroptosis coincident with cell cycle arrest. Cell Chem Biol. 2024 Feb 15;31(2):234-248.e13. doi: 10.1016/j.chembiol.2023.10.011. Epub 2023 Nov 13. PMID: 37963466; PMCID: PMC10925838.
Comments 5 : They wrongly mention that their 3D cultures are spontaneous spheroids since the low adhesion conditions induce the formation of these spheroids. The use of low adhesion conditions with medium without differentiation factors is a method for the enrichment of stem cells, which are effectively more resistant to different treatments. This is not the case since they only used low-adhesion dishes. What is the advantage of only using low adhesion?
Response 5 : The term “spontaneous” has been removed from the manuscript, and we have included a reference to acknowledge the relevance of using low-attachment plates for spheroid generation. (line 358-359) This method was specifically chosen to avoid the use of Matrigel or differentiation factors, thereby minimising potential confounding variables in treatment response in comparison to 2D-cultured cells. Since ferroptosis is a metabolic form of cell death, altering metabolic conditions with the addition of differentiation factors could interfere with the cell death process, whereas using different electrostatic forces helps mitigating this issue.
Comments 6 : The results should be grouped according to cell type and differential responses. As they are now, this is confusing and irrelevant since all cell lines respond practically the same way.
Response 6 : We understand the reviewer's concern; however, presenting the results grouped by cell type would be overly detailed and repetitive. Instead, we have chosen a methodological approach to streamline the presentation. Moreover, the fact that all cell lines respond in the same way reinforces the fact that JA has a broad range of targets. However, we have adapted figure 1E and made some adjustments to figures 1F, 3C and 3D to make it less confusing.
Comments 7 : Discussion: Extremely repetitive and speculative. This study cannot support a section discussing side effects and immune modulation of the microenvironment.
Response 7 : The discussion has been restructured and condensed to better align with the scope of our manuscript, ensuring a more focused and evidence-based presentation. (line 292-328)
Reviewer 3 Report
Comments and Suggestions for Authors
The work by Cuvelier et al. is excellent and very well constructed. It's particularly appreciable that the authors have carried out this work on numerous different-stage 2D breast cancer cell lines. Then, the transposition of the results obtained on spheroids and organoids proves very convincing.
Without reservation, it seems to me that this study can be published directly in IJMS.
Author Response
We would like to express our sincere gratitude for your review. We are pleased to hear that you found our work satisfactory and appreciate your positive feedback. Your support is truly encouraging and motivates us to continue our efforts in this direction.
Thank you once again for your time and consideration.
Sincerely,
Géraldine Cuvelier
Round 2
Reviewer 1 Report
Comments and Suggestions for Authors
Regarding response 2, the legend does not correspond to new fig 1B anymore. Please clarify
Regarding response 3, the added sentence (in italics below) is still unclear, please rephrase (Each symbol represents an independent replicate, with each cell line assigned a unique symbol)
Author Response
We thank the reviewer for its valuable comments. We have carefully addressed the points raised and incorporated the necessary revisions accordingly. Thank you for your time and insightful feedback.
Comments 1: Regarding response 2, the legend does not correspond to new fig 1B anymore. Please clarify
Response 1: Figure 1B has been updated to accurately correspond to the figure legend. (line 153)
Comments 2: Regarding response 3, the added sentence (in italics below) is still unclear, please rephrase (Each symbol represents an independent replicate, with each cell line assigned a unique symbol)
Response 2: Regarding comment 3, we have revised the sentence for clarity. The new wording is:
"Each symbol represents a distinct biological replicate, each cell line being identified by a symbol of a unique shape." (line 222-223 and 225-227)
Reviewer 2 Report
Comments and Suggestions for Authors
is accepted in this new version